# OpenReview forum: "Scaling LLM Test-Time Compute Optimally Can be More Effective than Scaling Parameters for Reasoning"
_ICLR.cc/2025/Conference — ICLR 2025 Oral_

### Official Review · Reviewer_u4qM · 2024-11-01

**Soundness:** 4
**Presentation:** 4
**Contribution:** 3
**Rating:** 8
**Confidence:** 4

**Summary:**

The authors are investigating the relationships between scaling inference-time compute and training-time compute. With that said, the paper first investigates and characterizes different types of inference-time compute. There are two main reasons: the first is to change the LLM proposal distribution by having an additional set of tokens, and the second is to use a verifier to rank the best response. Building on top of this, they try to unify those two inference-time compute strategies together, and create a compute-optimal inference-time strategy that's conditioned on question/prompt (more specifically, question difficulties). Lastly they draw comparison to training-time scaling, and find that it is more beneficial to scale inference-time for easier questions.

**Strengths:**

- The unification of inference-time computing methodologies is useful. There are a sea of inference-time compute strategies, and the unification of this makes the discussion of it much easier. Although I have concerns that some other papers talk about inference-time compute method categorization, for details, see Question (1) below. This is not major, but I wish the authors can address and clarify.
- This unification also makes the experiments fairly comprehensive and thus having more confidence in the results.
- Trying to do a fair comparison between inference-time scaling and training-time scaling is novel. I think the authors have framed a good problem and showed a nice analysis.

**Weaknesses:**

- I partially disagree with the claim that the authors made at the end of the introduction. I think what the authors have shown doesn't mean scaling test-time computing can be preferable to scaling pretraining compute. For both medium and hard questions, using the same compute to scale pertaining works much more in favor of scaling inference compute. The advantage only comes in for easier questions, which I would argue is less important. Plus one can always do inference scaling for larger models. I think this claim may need to be justified more.
- When training a verifier, is it more fair to include the compute to train the verifier as part of the inference-time compute? Similarly for the fine-tuned revision model.
- It is a bit difficult to fathom what "compute optimal" is exactly. How is this obtained or how is it optimized? I understand that strategies are selected based on question difficulties but providing the exact detail would be nice.
- The separation of Verifier and Revision is a bit confusing as both require a PRM. The main distinction I think between sections 5 and 6 is one is using search and another is using revision.

**Questions:**

- Both [1] and [2] talk about the categorization of scaling inference-time compute. [1] separates inference compute into proposer and evaluator which is fairly similar to what this paper has proposed. [2] uses a more fine-grained definition to demonstrate different techniques. I   think both of them should be cited and discussed accordingly.
- Figure 3 (left) is a bit hard to interpret. What does each color mean?
Writing and Styling
- Line 076 "We find that...". The wording is a bit awkward.

[1] Wang et al.,  Reasoning in Token Economies: Budget-Aware Evaluation of LLM Reasoning Strategies

[2] Saad-Falcon et al., Archon: An Architecture Search Framework for Inference-Time Techniques

---

> ### Author Response · Authors · 2024-11-19
> **Rebuttal part 1 of 2**
>
> We thank you for taking the time to review our paper. We are glad you like our paper. To address your concerns, we have: clarified the writing further in places you expressed concern and answered each of your questions. Please let us know if anything is still unclear or if you have further questions. We are happy to clarify anything! If your concerns are addressed, we would appreciate it if you would be willing to raise your score. We respond to each of your main concerns below:
>
> > I partially disagree with the claim that the authors made at the end of the introduction. I think what the authors have shown doesn't mean scaling test-time computing can be preferable to scaling pretraining compute. For both medium and hard questions, using …
>
> We do not view our analysis as the last word on test-time scaling, but rather as merely one of the first steps, in which we demonstrate the efficacy of a set of fairly straightforward techniques from the existing literature. While we believe that our statement is correct and that test-time compute can be preferable to pretraining in certain settings (either with easy/medium questions or a low inference budget), we can add additional qualification about potential drawbacks and nuances. There is significant room for future work to improve upon our methods, as discussed in Appendix A. We also further qualify this statement further in the updated draft.
>
> > When training a verifier, is it more fair to include the compute to train the verifier as part of the inference-time compute? Similarly for the fine-tuned revision model.
>
> In our case the compute needed for finetuning is trivial relative to pretraining (e.g. less than one thousandth) and thus is has no meaningful impact on the conclusions of our analysis. However, future work may desire to scale up these techniques to training on orders of magnitude more problems during finetuning (either on the same task, or on multiple tasks as you suggested) than the 12k problems from the MATH training set that we used in this work, in this case finetuning compute may become significant. Our goal is not to demonstrate the strongest possible results with test-time compute, but rather to demonstrate what is possible to start with. We therefore leave questions of scaling finetuning for test-time compute as future work. However, we include a note about this as an avenue for future work in the discussion section of the updated draft.
>
> > It is a bit difficult to fathom what "compute optimal" is exactly. How is this obtained or how is it optimized? I understand that strategies are selected based on question difficulties but providing the exact detail would be nice.
>
>
> Compute-optimal scaling refers to running an inference-time method with the best hyperparameters, _specific_ to a given input. That is, given a compute budget B, the compute optimal strategy prescribes a set of hyperparameters customized to a given question so as to obtain the best performance within this budget B. As an example of implementing compute optimal scaling for allocating compute between sequential revisions and parallel best-of-N for a given question we: 1) compute the performance of both techniques on all difficulty bins using a held-out validation set; 2) given a new test question, estimate it’s difficulty and then deploy the strategy which had the highest performance within that difficulty bin on the validation set. We include this example in Section 3.2 of the updated draft, and in the “compute-optimal scaling” part of Sections 5 and 6, we include additional details of the specific hyperparameters that we optimize over. Please let us know if this is still unclear or if you have further questions. We are happy to clarify anything.
>
> > The separation of Verifier and Revision is a bit confusing as both require a PRM. The main distinction I think between sections 5 and 6 is one is using search and another is using revision.
>
> While you are right that both verification and revisions use a PRM, revisions modifies the base distribution (i.e., base LLM as well) whereas search using a PRM uses the base LLM directly as the proposal distribution. Concretely, in Section 5, we fix the base LM as the proposal distribution and focus our experiments around search with a learned verifier. In Section 6, we instead focus on optimizing the proposer by training it to do revisions. While there is also a verifier employed in Section 6, the experiments are not centered around different search techniques. We add additional clarifications about this to the updated draft. Do let us know if this explanation helps to clear things up.

---

> > ### Author Response · Authors · 2024-11-19
> > **Rebuttal part 2 of 2**
> >
> > > Both [1] and [2] talk about the categorization of scaling inference-time compute. [1] separates inference compute into proposer and evaluator which is fairly similar to what this paper has proposed. [2] uses a more fine-grained definition to demonstrate different techniques. I think both of them should be cited and discussed accordingly.
> >
> >
> > We have added both of these citations to the updated draft. We note that our work differs methodologically from [1] in that we do not conduct test-time scaling by merely chaining together prompts, but rather we finetune models specifically for these capabilities, and in fact prior work shows this finetuning is needed (e.g., see RISE, Qu et al. NeurIPS 2024, which shows that finetuning to needed to get positive revision performance and prompting very strong models is also not enough). [1] also shows that prompting models for test-time scaling generally underperforms simple baselines like majority voting. This is a point made by other related works [3] which we also cite in our paper, and this finding was part of our original motivation for conducting this analysis: to better understand how test-time compute scaling behaves after models have been finetuned for the task. [2] is an interesting concurrent work to ours which studies how to chain together multiple modules for optimizing test-time scaling. However, their analysis is focused on chaining prompts together for test-time scaling, rather than enabling better use of test-time computation via finetuning.
> >
> > > Figure 3 (left) is a bit hard to interpret. What does each color mean? Writing and Styling
> >
> > Good catch! We correct this in the updated draft. To answer your question here:
> > green=majority
> > orange=best-of-N
> > blue=beam-search
> >
> > > Line 076 "We find that...". The wording is a bit awkward.
> >
> > Fixed!
> >
> > [3]: [Large Language Models Cannot Self-Correct Reasoning Yet](https://arxiv.org/abs/2310.01798)

---

> > > ### Author Response · Authors · 2024-11-24
> > >
> > > Dear Reviewer u4qM,
> > >
> > > We hope you had a chance to go over our responses and revisions which clarified the writing further in places you expressed concern and answered each of your questions. Since the discussion period is ending in two days, please let us know if you have any comments or remaining concerns. Thanks a lot, we are happy to discuss further.
> > >
> > > Best,
> > > Authors

---

### Official Review · Reviewer_hoB2 · 2024-11-02

**Soundness:** 4
**Presentation:** 3
**Contribution:** 4
**Rating:** 8
**Confidence:** 5

**Summary:**

This paper investigates the best way to scale test-time computation in LLMs to improve their performance on challenging tasks (MATH). The authors analyze two primary mechanisms: (1) searching against dense, process-based verifier reward models, and (2) updating the model's proposal distribution over responses through sequential revision. They find that the effectiveness of different test-time compute strategies varies significantly based on question difficulty. Using this insight, they propose a "compute-optimal" scaling strategy that adaptively allocates test-time compute per question, achieving 4× better efficiency compared to best-of-N baselines. In a FLOPs-matched evaluation, the authors demonstrate that for problems where a smaller base model shows some success, test-time compute can outperform a 14× larger model.

**Strengths:**

1. The paper provides a unified perspective on test-time compute, namely modifying either the proposal distribution or searching with an external verifier. This framework helps systematically understand and analyze different methods.

2. The authors conduct comprehensive experiments on the MATH dataset using PaLM-2 models, comparing different test-time compute strategies across varying problem difficulties. The analysis is thorough, with detailed ablation studies showing the effectiveness of both mechanisms they proposed and the trade-offs between different approaches.

3. The paper's findings have significant practical implications for model deployment. The demonstration that test-time compute can sometimes substitute for model size suggests a path toward deploying smaller, more efficient models while maintaining performance through clever use of inference-time computation.

4. The paper provides very insightful analysis on searching with PRMs and iterative self-refinemet, both of which have drawn lots of attention from researchers but still lack in-depth understanding. I pretty much enjoy the detailed analysis on the three ways of use of a PRM.

**Weaknesses:**

1. All experiments are done solely on the MATH dataset, which not only raises concerns on the genralizability of the conclusion, but also seems unfair when compared to scale up training time compute. Given multiple downstream datasets, we only need to scale up training time compute for once while have to scale up inference overhead every time. In this case, if we match the total test time compute with training time one, each task will be allocated much fewer inference time budget, which may lead to very different conclusions.

2. When scaling inference time compute, dedicated efforts are needed to improve the verifier performance or models' capability of self-correction, which may also consume a non-trivial proportion of compute. This may also lead to unfair comparison to training time compute scaling.

3. Most presentations are excelllent but I there are still some minor issues. was a bit confused when reading Figure 3 (Left) since there is no legends and the overlap of bars make colors even more complex. Also, the main body oof the paper should be self-constrained, so it may not be quite suitable to put the results in Sec 5.3 to Figure 14 which is inconvenient to readers.

**Questions:**

- It seems all PRM results are presented in the way of *weighted* Best-of-N. Is PRM alone not working? If so, is it fair to compare lookahead search with the other two, given that the value in lookahead search can not be augmneted by majority vote?

---

> ### Author Response · Authors · 2024-11-19
> **Rebuttal**
>
> We thank you for taking the time to review our paper and are glad for the positive review. We appreciate that you find our experiments to be comprehensive and our analysis to be insightful. We respond to each of your main concerns below:
>
>
> > Given multiple downstream datasets, we only need to scale up training time compute for once while have to scale up inference overhead every time. In this case, if we match the total test time compute with training time one, each task will be allocated much fewer inference time budget, which may lead to very different conclusions.
>
> > When scaling inference time compute, dedicated efforts are needed to improve the verifier performance or models' capability of self-correction, which may also consume a non-trivial proportion of compute. This may also lead to unfair comparison to training time compute scaling.
>
> There are great points! In our case the compute needed for finetuning is trivial relative to pretraining (e.g. less than one thousandth) and thus is has no meaningful impact on the conclusions of our analysis. However, future work may desire to scale up these techniques to training on orders of magnitude more problems during finetuning (either on the same task, or on multiple tasks as you suggest) than the 12k problems from the MATH training set that we used in this work, in this case finetuning compute may become significant. Our goal was not to demonstrate the strongest possible results with a test-time compute, but rather to demonstrate what is possible using a few straightforward ideas from the existing literature. We therefore leave questions of how test-time compute scaling improves with additional finetuning data/tasks as future work. However, we include a note about this as an avenue for future work in the discussion section of the updated draft.
>
> > Most presentations are excellent but I think there are still some minor issues. was a bit confused when reading Figure 3 (Left) since there is no legends and the overlap of bars make colors even more complex
>
> Glad that you find most presentations excellent. Good catch regarding Figure 3! We correct this in the updated draft. To answer your question here:
> green=majority
> orange=best-of-N
> blue=beam-search
>
> > Also, the main body oof the paper should be self-constrained, so it may not be quite suitable to put the results in Sec 5.3 to Figure 14 which is inconvenient to readers.
>
> We have moved this figure into the main paper in the updated draft.
>
> > It seems all PRM results are presented in the way of weighted Best-of-N. Is PRM alone not working? If so, is it fair to compare lookahead search with the other two, given that the value in lookahead search can not be augmented by majority vote?
>
> Best-of-N performs well; it just somewhat underperforms weighted best-of-N. Specifically, best-of-N refers to selecting the best single response from all samples. best-of-N weighted effectively combines best-of-N and majority voting by summing over the scores of all solutions which reach the same final answer and returning the answer with the greatest sum. We include a plot with Best-of-N in appendix G.2 of the updated draft.

---

> > ### Comment · Reviewer_hoB2 · 2024-11-21
> > **Respond to Rebuttal**
> >
> > Thanks for your response. Is it possible that your PRM outperforms major vote simply because you used a weak model (with major vote ending up achieving an acc of 30). Accoridng to fig. 7 of [1], stronger models will have higher major voting performance and sometimes can outperform best-of-N or even weighted best-of-N. If so, is your proposed verifier-based methods really a good solution to scale up inference compute?
> >
> > [1] Large Language Monkeys: Scaling Inference Compute with Repeated Sampling. Brown et al. 2024

---

> ### Author Response · Authors · 2024-11-21
> **Majority Vote Response**
>
> Thanks for the response and for the great question! Regarding the majority voting baseline in our results, we do not believe that our PRM outperforms majority vote simply because we used a weak model. This is because we believe this result is consistent with other results in the community:
>
> * Prior results show that verifiers outperforming majority voting is a standard result, including in cases where much larger or stronger models are used. For example, the MathShepard paper [1] shows in Table 1 that the MathShepard verifier beats majority vote for a number of strong base LMs (DeepSeek-67B, LLemma-34B, LLaMA 2-70B). Additionally, the Let’s Verify Step by Step paper [2] shows a PRM verifier significantly outperforming majority vote with a very strong base model in Figure 3 (majority is ~68%).
> * In our results, we see a very consistent trend, across three different base LLMs in Figure 3 (right) and Figure 16, in which majority voting performs similarly to ORM/PRM on easy questions, but on medium/hard questions the gap widens significantly with verifiers performing much better. Indeed, for a much more capable model with much higher absolute performance on the MATH benchmark (say 90-95% for example), all problems in the benchmark would effectively become “easy problems” and thus majority voting may be all we need. However, even for such a model we should still be able to construct much harder questions than anything in MATH (e.g. IMO level), which would benefit from a verifier. There should always be “hard problems” (unless all reasoning problems can be solved), just that they may not exist in the MATH dataset.
>
> For these reasons, we have a strong reason to believe that the general findings that we observe on MATH in our setting should hold more generally, even as we move to much more capable LMs. That said, we want to clarify that we are not proposing that PRMs are the best way to use test-time computation -- there are likely better ways of using test-time computation that lead to better results. Our point is just to show that using test-time compute via PRMs **can** enable us to obtain better performance, though it may not always the best way to do so.
>
> [1]: [Math-Shepherd: Verify and Reinforce LLMs Step-by-step without Human Annotations](https://arxiv.org/abs/2312.08935)
>
> [2]: [Let’s Verify Step by Step](https://arxiv.org/abs/2305.20050)

---

> > ### Comment · Reviewer_hoB2 · 2024-11-25
> > **Respond to Author Response**
> >
> > Thanks for your response. However, the figure 5 in Math-Shepherd paper does support my thought though. But in general, I agree with the view that the main purpose of this paper is to showcase the power of inference scaling, even though the cases in this paper seem not be the optimal. I will keep my scores.

---

### Official Review · Reviewer_MYbX · 2024-11-02

**Soundness:** 3
**Presentation:** 3
**Contribution:** 3
**Rating:** 6
**Confidence:** 3

**Summary:**

This paper target the challenges in effectively utilizing additional computation at test time to improve the accuracy of their responses, particularly for complex tasks. This is important to explore how to tradeoff inference-time and pre-training compute. This paper trying to understand the scaling behaviors of test-time inference methods. This work analyze main mechanisms, and observed that the effectiveness of recent methods varies depending on the specific problem and the base LLM used.The observation motivates applying a “compute-optimal” scaling strategy, which acts to most effectively allocate test time compute adaptively per prompt. This approach selects the most effective method for utilizing additional computation based on the specific prompt and question difficulty.

**Strengths:**

The problem addressed in this paper is highly meaningful. This paper provide systematic analysis of different approaches for scaling test-time computes in LLMs. The observations could inspire further research and is beneficial for advancing the entire field.

The paper is well written and easy to follow.

**Weaknesses:**

1. Both the PRM and ORM models utilize the PaLM 2-S* base language model, which is costly. However, the computational cost associated with PRM during test time is not clearly defined and calculated.

2. In a specific domain, analyzing the trade-off between pre-training and test-time computation is more meaningful, but the paper primarily focuses on pre-training. As mentioned in the paper, "we expect test-time compute to be most helpful when models already have all the basic “knowledge” needed to answer a query, and instead the primary challenge is about drawing (complex) inferences from this knowledge." Pre-training can be leveraged to learn the basic "knowledge," while fine-tuning might be more effective in teaching the model how to draw complex inferences. Additionally, fine-tuning can be more efficient, providing greater benefits for the same computational cost.

3. Whether the observations are consistent across different datasets and models remains unclear.

4. I think some details are unclear, which may affect the strength of the observations. Since the main contribution of this paper is conducting analysis using methods proposed in previous works, clarifying these details is important.

**Questions:**

1. What do the different colors in Figure 3 (left) represent? It appears the legend is missing.

2. What is the accuracy of the difficulty prediction? If it is quite accurate, the results with model-predicted difficulty may not be compelling because they rely on a powerful model for estimating difficulty and do not account for the associated computational cost, which is unrealistic. In real-world scenarios, using a powerful model entails significant costs, while employing a less powerful model may result in inaccurate difficulty predictions. Therefore, it would be valuable to demonstrate whether the observation holds true with a less powerful difficulty estimation model.

3. What is the accuracy of the verifier? Does the performance of the verifier influence the observations?

4. Is the accuracy of the verifier consistent across different difficulty levels? The observations might be influenced by the accuracy of the verifier.

5. How many test questions are there in each difficulty bin? Is it consistently 100 questions per bin?

---

> ### Author Response · Authors · 2024-11-19
> **Rebuttal part 1 of 2**
>
> Thank you for the review and for the constructive feedback! We are glad that you liked the paper. To address your concerns, we have: 1) adjusted for the cost of calling the verifier in our analysis; 2) added additional results using other base models for PRM search; and 3) answered each of your questions. **Please let us know if our responses and edits to the paper address your concerns and if so, we would be grateful if you are willing to raise your score. We would be happy to discuss further.**
>
>
> > Both the PRM and ORM models utilize the PaLM 2-S* base language model, which is costly. However, the computational cost associated with PRM during test time is not clearly defined and calculated.
>
> Thanks for the question! We first note that these changes do not change the main conclusions of our analysis (we discuss why below), but we have now updated the draft to modify our analysis to account for the verifier cost. Using a verifier is twice as expensive as the majority voting baseline, so to account for this, we shift our best-of-N plots accordingly. Additionally, we can update our analysis of exchanging pre training to reflect the cost of calling the verifier. In practice, the effect of this is to half the R values (D_inference/D_pretrain) corresponding to each of the vertical lines in Figure 7. With these adjustments, the majority curves still fall below our best-of-N baselines and our conclusions with regards to how pre-training/test-time compute can be exchanged remain unchanged.
>
> > In a specific domain, analyzing the trade-off between pre-training and test-time computation is more meaningful, but the paper primarily focuses on pre-training.
>
> We might be misunderstanding your question, so please point out if this answer does not address your concerns. To our understanding, this question is about potentially analyzing tradeoffs of fine-tuning vs only using test-time computation. It is true that for our analysis in Section 7 we could have fine tuned the larger model as another baseline comparison. However, we do not believe this would represent a meaningful improvement in performance so as to change any of our main findings. Namely, finetuning models using techniques like Rest^EM or STaR on the same number of questions we use for training the verifiers/revision models (e.g. 12K questions from MATH) does not improve accuracy by more than ~5% on MATH [1]. Our test-time compute techniques instead improve performance by up to 30% in some cases. Therefore, we do not believe this would represent a meaningful improvement in performance so as to change any of our main findings.
>
> > Whether the observations are consistent across different datasets and models remains unclear…
>
> We have some additional results with training different size verifier models, and we see the same trends for these other models. We include these results in Appendix I of the updated draft.
>
> As for datasets, MATH is a standard dataset used in a number of related works [3,4,5], and it is really the only datasets which is hard enough, but not too hard, to provide a good test bed for analyzing test-time compute at our level of model capability. So we believe that the dataset should not be grounds for rejection. That said, if the reviewer has thoughts on what could be a better dataset to study test-time compute, we could consider incorporating that for the camera ready as well.
>
> > What do the different colors in Figure 3 (left) represent? It appears the legend is missing.
>
> Good catch! We correct this in the updated draft. To answer your question here:
> green=majority
> orange=best-of-N
> blue=beam-search

---

> > ### Author Response · Authors · 2024-11-19
> > **Rebuttal part 2 of 2**
> >
> > > What is the accuracy of the difficulty prediction? If it is quite accurate, the results with model-predicted difficulty may not be compelling because they rely on a powerful model for estimating difficulty and do not account for the associated computational cost, which is unrealistic. In real-world scenarios, using a powerful model entails significant costs, while employing a less powerful model may result in inaccurate difficulty predictions. Therefore, it would be valuable to demonstrate whether the observation holds true with a less powerful difficulty estimation model.
> >
> > This is a great question! Our results do not require the difficulty bins to agree with the oracle difficulty bins accurately. In fact, in our experiments, predicted difficulty bins agree with the oracle difficulty bins 52.8% of the time (random would be close to 20%). We also report the precision of our predicted bins relative to the oracle for each bin:
> >
> > Level 1: 0.77
> >
> > Level 2: 0.48
> >
> > Level 3: 0.41
> >
> > Level 4: 0.38
> >
> > Level 5: 0.6
> >
> > This means that we do not require difficulty prediction to be very accurate. Moreover, recent work from [2], demonstrates that cheaply evaluating question difficulty can be very effective. Our results serve to demonstrate what is possible with scaling test-time computation using a handful of simple approaches. Encouraging results from recent work like [2] suggest that these findings can be made practically useful.
> >
> > We also want to remark that difficulty is merely a statistic for choosing the right hyperparameters for allocating inference-time compute. In general, we are not suggesting that difficulty is the best statistic for allocating compute, but it shows us that it is possible to allocate compute with _some_ statistic specific to a question that leads to better usage of inference-time compute. This statistic does not need to align with human notions of difficulty in general.
> >
> >
> > > What is the accuracy of the verifier? Does the performance of the verifier influence the observations?
> >
> > Our verifier achieves 91.4% binary classification accuracy on the MATH-500 test-set. We would expect the verifier accuracy to influence our findings. For instance, a random-uniform verifier is expected to have very poor test-time scaling. Similarly if we could train a verifier on millions of problems, we would expect it to scale well beyond what we study in this paper. Such a verifier would likely show much better test-time scaling.
> >
> > We believe that our verifier is well-trained because it shows better test-time scaling than the verifier used in the Math Shepard paper [4] (e.g. in Figure 14 we see that our verifier scales from ~10% to ~40% with no signs of a plateau, whereas the math shepard paper in Figure 3 scales from ~30% to ~45% and begins to show signs of saturation around 45%), which introduced the process supervision training technique that we use in our paper.
> >
> >
> > > Is the accuracy of the verifier consistent across different difficulty levels? The observations might be influenced by the accuracy of the verifier.
> >
> > Indeed the verifier accuracy is a bit lower at the higher difficulty levels. Specifically we get classification accuracies at each difficulty level (in order of increasing difficulty):
> >
> > Level 1: 99.7%
> >
> > Level 2: 97.8%
> >
> > Level 3: 94.1%
> >
> > Level 4: 86.3%
> >
> > Level 5: 82.4%
> >
> > This should not be a surprise, since our difficulty bins are determined by the base model’s pass@1 on a given question and the verifier is finetuned from this very same base model, we should expect it to struggle with similar questions. We also defer to our answer to the previous question: we believe our verifier is well trained, but there are many ways we could also improve our verifier and we leave this exploration to future work. Our goal is to demonstrate what is  possible with test-time compute using a set of relatively straightforward techniques from the existing literature.
> >
> > > How many test questions are there in each difficulty bin? Is it consistently 100 questions per bin?
> >
> > Yes, 100 questions per bin, with one exception: level 4 of our oracle bins has 105 questions and level 5 has 95 questions. This imbalance is just due to a boundary condition/ties in the quantile computation, causing one bin to inherit slightly more questions than the other. We have clarified this in the paper now.
> >
> >
> >
> > [1]: [Beyond Human Data: Scaling Self-Training for Problem-Solving with Language Models](https://arxiv.org/abs/2312.06585)
> >
> > [2]: [Learning How Hard to Think: Input-Adaptive Allocation of LM Computation](https://arxiv.org/abs/2410.04707)
> >
> > [3]: [RL on Incorrect Synthetic Data Scales the Efficiency of LLM Math Reasoning by Eight-Fold](https://arxiv.org/abs/2406.14532)
> >
> > [4]: [Math-Shepherd: Verify and Reinforce LLMs Step-by-step without Human Annotations](https://arxiv.org/abs/2312.08935)
> >
> > [5]: [Let's Verify Step by Step](https://arxiv.org/abs/2305.20050)

---

> > ### Comment · Reviewer_MYbX · 2024-11-21
> >
> > Thank you for your response. Most of my questions have been answered. For the second weakness, yes, I want to ask for fine-tuning instead of pertaining; thank you for answering it.
> >
> > However, I still have concerns about whether those observations can be generalized to datasets from different domains and difficulty levels. Reviewer 19PN hoB2 also mentioned a similar concern.
> >
> > As you said, MATH is hard enough, but not too hard. Then what will happen if the dataset is pretty hard or the generation model is less powerful? What will happen if the dataset is easy or the LLM is super powerful, such as GPT4? Also, math is just one domain, what about other domains, such as MMLU, CSQA, Triviaqa? For example, Mistral-7B can only achieve about 10% accuracy on the NQ-Open dataset, and for HellaSwag and MMLU, the accuracy is about 60%. Are the observations consistent across those datasets? I'm not saying that experiments on all those kinds of models and datasets are needed, it's impossible to contain everything, but I think just MATH is insufficient to support all those conclusions.

---

> ### Comment · Reviewer_MYbX · 2024-11-21
>
> Regarding my questions, it seems the authors generally agree that the performance of test-time scaling is highly sensitive to the accuracy of the verifier. I believe it would be valuable to include a more detailed discussion on this point. I agree that the verifier is well trained, and I believe the observations based on a well-trained verifier. That's exactly the reason why I have concerns about the verifier. I'm curious what will happen if we can not get a well-performing verifier or if we can not afford high verifier training costs. Also, as the response 'Indeed the verifier accuracy is a bit lower at the higher difficulty levels.', what happens if it's more unbiased?
>
> Specifically, with different verifiers, the exchange between pretraining and test-time computing may vary significantly. It’s also unclear whether observations such as “beam search is more effective on harder questions and at lower compute budgets” remain consistent under different verifier configurations. Could you provide additional information or clarification on this?
>
> Also, thanks for the effort to add the verifier's inference cost. But should the verifier's training cost also be considered in the pretraining and test-time compute exchanging discussion?

---

> > ### Author Response · Authors · 2024-11-23
> > **Discussion on verifiers**
> >
> > > Regarding my questions, it seems the authors generally agree that the performance of test-time scaling is highly sensitive to the accuracy of the verifier. I believe it would be valuable to include a more detailed discussion on this point. I agree that the verifier is well trained, and I...
> >
> > > Specifically, with different verifiers, the exchange between pretraining and test-time computing may vary significantly. It’s also unclear whether observations such as “beam search is more effective on harder questions and at lower compute budgets” remain consistent under different verifier configurations. Could you provide additional information or clarification on this?
> >
> > > Also, thanks for the effort to add the verifier's inference cost. But should the verifier's training cost also be considered in the pretraining and test-time compute exchanging discussion?
> >
> > Thank you for the questions! We do agree that the exchange between pretraining and test-time compute will vary significantly depending upon the accuracy of the verifier, and there is no reason to believe that a worse verifier (e.g., one that is completely inaccurate) would lead to any meaningful test-time scaling behavior. Please do note that we never claimed in our paper or elsewhere that our observations should always hold with any verifier either, so we believe that the broader observation about the **existence** of an approach to scale up test-time compute in a question-dependent manner such that it outperforms scaling parameters should hold. Please let us know if any of our wording suggests otherwise, and we will happily clarify it to avoid misunderstanding.
> >
> > That said, **we also note that while our verifier is well trained, we do not believe that it is the best verifier one can obtain.** This is because the finetuning cost for training our verifier was trivial (less than one thousandth of pretraining), since it was only trained on the 12k problems in the MATH training split, and thus the finetuning cost has no material impact on our FLOPs analysis in Section 7. However, one could instead train a verifier model on millions of unique questions scraped from the web, and it would likely perform significantly better (albeit at a much higher finetuning cost). We think this is a really interesting question: how does test-time compute scaling improve with additional finetuning. However, we leave this to future work, since ***our goal is not to find the best way to use test-time computation*** -- there are likely better ways of using test-time computation that lead to better results, as we already remark in the paper. Our goal is just to show that there are ways to use test-time compute that can work well, not that PRMs or revisions are the best way to do so.
> >
> > **We now also provide a new analysis showing that our conclusions can in fact hold with a weaker verifier**. In the attached figure [link](https://imgur.com/a/4FfLqMH), we conduct our FLOPs analysis from Section 7, using best-of-N weighted with verifiers of varying quality levels:
> > 1) A: a “uniform random” verifier (a.k.a majority voting; BoN weighted with a random verifier converges to majority voting)
> > 2) B: our PRM verifier but with 20% iid label flip noise (so if the verifier predicts p, we flip it to 1-p 20% of the time)
> > 3) C: our standard PRM verifier without noise
> > 4) D: Majority voting using parallel samples from the revisions model with no PRM (i.e., a uniform random verifier on top of the revisions model)
> >
> > We see that while majority voting (A) generally underperforms the larger pretrained model, the noised verifier (B) can outperform pertaining on easy/medium questions in settings with low inference requirements. We also note that majority voting with a revisions model (D) can outperform scaling model parameters on easy and medium questions, without using a verifier suggesting **that it is still possible to get decent test-time scaling that outperforms pretraining with a less capable verifier (or no verifier, if the base model is finetuned in a certain way).** Of course, not any verifier will enable such a result, but this shows that it is possible. We will add this discussion to the paper.
> >
> > **To answer your question about the generality of the observation “beam search is more effective on harder questions and at lower compute budgets”**: as we noted in our original response, we include some additional results with training different size verifier models in Appendix I, and we see the same trends for these other models (namely, more sophisticated search methods like beam search help more on harder problems).
> >
> > **Please let us know if these responses address your questions. We are happy to discuss further. Thanks for engaging in a discussion with us.**

---

> > > ### Comment · Reviewer_MYbX · 2024-11-25
> > >
> > > Thank you for the detailed discussion. I will increase my score to 6.
> > >
> > > Overall, I find the main observations meaningful and believe they could inspire further research. However, I think the paper could be further improved with additional experiments, such as evaluations on other datasets and models. Additionally, some conclusions could be more rigorous. For instance, are the findings specific to reasoning problems, or do they generalize more broadly? Should there be restrictions on the verifier? Furthermore, some details are missing in the paper, particularly in the first version.

---

> ### Author Response · Authors · 2024-11-22
>
> > Thank you for your response. Most of my questions have been answered. For the second weakness, yes, I want to ask for fine-tuning instead of pertaining; thank you for answering it. However, I still have concerns about whether those observations can be generalized to datasets from different domains and difficulty levels. Reviewer 19PN hoB2 also mentioned a similar concern...
>
> We respond to this comment first, and we will respond to your other comment shortly.
>
> Thank you for the response! We’re glad that we were able to answer most of your questions. Regarding datasets, unfortunately most datasets traditionally used in the LM literature are not a great fit for studying test-time compute. For example MMLU and triviaQA are more knowledge intensive, rather than reasoning. In fact, according to [1], which conducted an extremely thorough meta-analysis of which tasks benefit from chain of thought, most of these datasets (CSQA, MMLU, etc..) do not benefit from chain of thought. Only pure math tasks saw improvement. This is not to say that we expect test-time/reasoning compute to only benefit on math, it just means that the settings where today’s generation of LMs actually see improvement from reasoning are limited. We expect reasoning will likely help in many settings long term, such as agentic problems. **We will consider other coding datasets (e.g., Codecontests) for the final version of the paper. However, according to [2] llama 3-8B-Instruct has a very low oracle pass@100 rate on Codecontests (e.g. 10-15% verses 80-90% on MATH) , so obtaining useful results on this task may have a very high time/computational cost, and thus it would be infeasible to get this running in the rebuttal period.**
>
> That said, we believe that despite that fact that our analysis is conducted only on MATH, our difficulty bin analysis does provide some insights about how we should expect different test time compute techniques to perform on tasks of varying difficulties. Namely, we see that on really easy tasks simple techniques like majority voting are likely sufficient. However, on harder tasks, more sophisticated techniques are often beneficial.
>
> [1]: [To CoT or not to CoT? Chain-of-thought helps mainly on math and symbolic reasoning](https://arxiv.org/abs/2409.12183)
>
> [2]: [Large Language Monkeys: Scaling Inference Compute with Repeated Sampling](https://arxiv.org/abs/2407.21787)

---

### Official Review · Reviewer_19PN · 2024-11-10

**Soundness:** 3
**Presentation:** 3
**Contribution:** 4
**Rating:** 8
**Confidence:** 4

**Summary:**

The paper carries out comprehensive study on test-time compute. The authors study the topic from the following perspectives:
1. The first question they study is "How to scale test-time computation optimally". The authors propose an adaptive "computer-optimal" strategy for scaling up test-time compute. They evaluate the method against test-time inference methods, e.g., searching against dense, process-based verifier reward models, and updating the model's distribution over a response adaptively, given the prompt at test time. They study how to optimally combine these methods to reduce the test-time budget and achieve better performance.
2. Then they study to what extent test-time computation can substitute for additional pre-training, by comparing test-time inference on a smaller model compared to pre-training a ~14x larger model, under a FLOPs-matched comparison.

Finally, the authors summarize their results with a finding that states *"in some settings it is more efficient to pretrain smaller models with less compute, and then apply test-time compute to improve model outputs"*.

**Strengths:**

1. The paper is fluent and coherent. The authors have clearly made their points so that the ideas they propose can easily be followed.
2. The paper carries out comprehensive experiments and analysis on the research questions they study, i.e. 1) scaling test-time compute optimally and 2) exchanging pre-training and test-time compute.
3. The experiments are sound.
- For the first question, the baseline they choose and develop over, e.g. best-of-n and revision, are strong, which makes their points more convincing. And the study on sequential to parallel ratio is also helpful for readers to better understand the merits of different test-time inference methods and how they contributes to the "compute-optimal" strategy.
- For the second question, the setup for FLOPs matched comparison is plausible and the ablations on model size and question difficulties are insightful, making their conclusions convincing and easy to understand.

**Weaknesses:**

1. Yet the adaptive "compute-optimal" strategy achieves great performance over other test-time baselines and can be used to substitute for pre-training in some cases, for me it's more like a proof-of-concept.
- The experiments are all on MATH and
- there're still many hyper-parameters under-explored, e.g.,
  - different verifiers;
  - different design of difficulty bins (number of bins and how to estimate difficulties),

along with the ones that have been explored in the paper:
  - types of test-time methods to select from (my understanding is we can have a set of many test-time methods to adaptively select from)
  - test-time compute need for different methods

It'll be much clearer and more helpful if the authors can discuss more about the possible "hyper-parameters", compensating the framework described in Eq. 1. (I see some of them are in the appendix, e.g. design of difficulty bins, but it would be much clearer for me if they can be put and discussed at the same place)

**Questions:**

1. My understanding from the paper is that different FLOPs budgets of SFT can also be taken into consideration, and we may be able to teach LLMs better specialized skills (e.g., revision) with more deliberate post-training. I don't see much ablations on the revision SFT in the paper. Do you think that could be the case?
2. This is an interesting and insightful paper. I have another question (better called thought) -- what if we scale on both test-time compute and pre-training compute, would there be another optimal solution under FLOPs-matched comparison? The paper only focuses on 1-to-1 exchange (scaling test-time compute only) but maybe that would also an interesting topic.

---

> ### Author Response · Authors · 2024-11-19
> **Rebuttal part 1 of 2**
>
> Thank you for the review and for a positive assessment of our paper. We are glad you appreciate the comprehensiveness of our experiments. We respond to each of your main comments and concerns below:
>
> > **Regarding the “proof-of-concept” nature of our analysis.**
>
> As stated in the introduction our findings “[suggest] that with a fairly naive methodology, scaling up test-time computation can already be preferable to pretraining, with only more improvements as test-time strategies mature.” We do not wish to claim that our work is the final word on test-time compute and indeed believe that there are many future directions to expand upon our work, including: alternative designs for difficulty bins, novel approaches to training verifiers, and fundamentally different methods for utilizing test-time computation. We could not study all of these due to computational and infrastructural limitations, but we discuss many of these possible directions in the discussion section (Appendix A). We believe that the specific experimental decisions we made in the paper provide for a strong starting-point for understanding test-time scaling (e.g. we use 5 difficulty bins as it aligns with the 5 bins in the MATH dataset; we study improving the proposer and verifier as these represent two primary axes with which test-time compute can be scaled as discussed in the paper; the specific methods we implement are not novel and instead based on existing approaches in the literature). We also note that there are some specific limitations to our work, such as the cost of estimating question difficulty (see limitations in Section A).
>
> In the specific case of our difficulty-bin analysis, our main contribution is to show the existence of a statistic that can help us predict how to best use test-time compute, since, in principle, no one test-time strategy is going to work on all problems (e.g., it is easy to construct worst case examples where each strategy is bad). Of course, while we use a fairly naive way of computing this statistic, recent work [1], already builds a cheap method to estimate this quantity, showing positive results.
>
> > It'll be much clearer and more helpful if the authors can discuss more about the possible "hyper-parameters", compensating the framework described in Eq. 1.
>
> Section 3.2 of the updated draft includes a clear example of how Eq. 1 is used. We also describe the specific hyperparameters considered in the compute optimal results sections of both Section 5 and 6 (the changes are in blue).

---

> ### Author Response · Authors · 2024-11-19
> **Rebuttal part 2 of 2**
>
> > My understanding from the paper is that different FLOPs budgets of SFT can also be taken into consideration
>
>
> The finite nature of math finetuning data (e.g. 12k questions in the MATH training set) means that the total SFT compute in terms of FLOPs contribute negligibly w.r.t pretraining FLOPs in Section 7 (to be precise, less than one-thousandth the compute used for pretraining) and thus will have no meaningful effect on our conclusions. We can spend additional FLOPs on each question by training for additional epochs or generating additional solution traces per questions but scaling in this way will quickly overfit [2] (plus we already do early stopping with all of our models).
>
> On the other hand, future works could consider experimenting with a much larger custom scraped dataset of, say hundreds of thousands of questions (or millions), to better understand the scaling of these different techniques with finetuning compute. It is likely that verifiers or revision models trained on this much larger dataset would perform much better. We leave this exploration to future work.
>
> > This is an interesting and insightful paper. I have another question (better called thought) -- what if we scale on both test-time compute and pre-training compute
>
>
> Thanks for the question. This is a very insightful point, there is certainly an interesting question of what is the optimal ratio of compute to allocate to pretraining versus test-time. Expanding upon our empirical findings, we hypothesize that the optimal model given a total FLOPs budget that accounts for pretraining and test-time compute will be different from the compute-optimal pre-training solution by itself. While we do not answer this question in the paper, it is an interesting question that future work should concretely answer, building on our analysis.
>
> To see intuition as to why the compute-optimal pre-training solution might change, consider over-training: while overtrained models are generally better when considering pretraining + standard inference (e.g. no extra test-time compute) FLOPs, they might be suboptimal when accounting for test-time FLOPs as these models may not produce samples with enough diversity for test-time search or revisions to be effective. We add a discussion of this to the discussion section in the updated draft.
>
>
> [1]: [Learning How Hard to Think: Input-Adaptive Allocation of LM Computation](https://arxiv.org/abs/2410.04707)
>
> [2]: [RL on Incorrect Synthetic Data Scales the Efficiency of LLM Math Reasoning by Eight-Fold](https://arxiv.org/abs/2406.14532)

---

> > ### Comment · Reviewer_19PN · 2024-11-20
> >
> > Thank you for your clarification and discussion. I do believe this paper is solid and will pave the way for future studies not only in test-time methods of LLMs (the first part of the paper), but also in how we exploit the full potential of LLMs with proper "continual" training (the second part of the paper). Based on that, I will keep my rating as 8 and raise my confidence from 3 to 4.

---

### Public Comment · ~Charlie_Victor_Snell1 · 2025-03-03
**Thank you + Additional pointers**

We would like to wholeheartedly thank the reviewers and the AC for their feedback on our paper. Your feedback has been extremely helpful to improve the quality and clarity of our work, and we are looking forward to presenting the work in Singapore!

We would like to highlight some of the changes we have made since the rebuttal period to further account for the reviewer’s concerns. Specifically, we have:

1) Adjusted the paper’s title to more precisely scope the contributions of our paper towards tasks which involve reasoning;

2) Added discussion of additional related work to better situate our work in the literature;

3) Included additional experiments in the appendix showing that our claims about test-time scaling hold even with weaker verifier models, effectively accounting for the effect of fine-tuning on our findings; and

4) Included, as part of our reproducibility statement, a pointer to some newer works that reproduce some of the main findings of our work for readers to build upon.

In particular, we would like to highlight point 4) above. Namely, since the paper submission deadline, two subsequent works [1,2] have independently reproduced our main findings using open LLMs (e.g. LLaMA and Qwen models) and using other math benchmarks (e.g. AIME-2024). These results provide additional evidence to support the title and main claim made in this paper that scaling test-time compute can outperform scaling model parameters.

[1]: [Scaling Test Time Compute with Open Models](https://huggingface.co/spaces/HuggingFaceH4/blogpost-scaling-test-time-compute)

[2]: [Can 1B LLM Surpass 405B LLM? Rethinking Compute-Optimal Test-Time Scaling](https://arxiv.org/abs/2502.06703)

---

### Meta-Review · Area_Chair_kfxy · 2024-12-21

**Metareview:**

The paper "Scaling Test-Time Compute Optimally Can be More Effective than Scaling LLM Parameters" investigates the potential of leveraging test-time computation in large language models (LLMs) as an alternative to scaling model parameters during pretraining. The authors explore two primary mechanisms for scaling test-time computation: (1) searching against dense verifier reward models, and (2) adaptively updating the model's response distribution based on the prompt. They introduce a "compute-optimal" strategy that dynamically allocates test-time computation based on the difficulty of the prompt, demonstrating significant performance improvements. In a FLOPs-matched evaluation, the study reveals that test-time computation can outperform a model 14 times larger on tasks where the smaller model has a non-trivial success rate. The experiments are conducted primarily on the MATH dataset, with implications for future LLM development and deployment strategies.

#### Contribution
1. **Compute-Optimal Strategy**: The paper proposes a novel approach to allocate test-time computation adaptively based on prompt difficulty, achieving up to 4x better efficiency than traditional best-of-N baselines.
2. **FLOPs-Matched Evaluation**: The authors provide a rigorous comparison showing that test-time computation can outperform significantly larger models in specific scenarios, highlighting a potential shift in how LLMs are optimized.
3. **Empirical Validation**: Comprehensive experiments on the MATH dataset demonstrate the efficacy of the proposed methods, providing a foundation for future research on test-time compute scaling.

#### Weaknesses
1. **Limited Dataset Scope**: The study's reliance on the MATH dataset raises concerns about the generalizability of the findings to other domains or datasets, which may not exhibit similar characteristics or difficulty distributions.
2. **Verifier Dependence**: The effectiveness of the proposed methods is heavily reliant on the accuracy of the verifier model, which may pose practical challenges in real-world applications due to training costs and domain specificity.
3. **Lack of Fine-Tuning Analysis**: The paper focuses predominantly on pretraining compute, potentially overlooking the benefits of fine-tuning, which could offer a more efficient alternative for specific tasks.
4. **Hyperparameter Exploration**: Some critical hyperparameters, such as verifier design and difficulty estimation methods, are not fully explored, limiting the robustness of the proposed framework.

**Additional Comments On Reviewer Discussion:**

**Points Raised by Reviewers and Authors' Responses:**

1. **Generalizability Concerns (MYbX, hoB2):**
   - **Concern**: The reliance on the MATH dataset limits the generalizability of the findings.
   - **Response**: The authors acknowledged the limitation and discussed the challenges of finding suitable datasets beyond MATH. They suggested exploring coding datasets like Codecontests in future work but noted the high computational cost as a barrier for the rebuttal period.
   - **Outcome**: The concern remains partially addressed, as the authors did not provide additional experimental evidence beyond MATH.

2. **Verifier Dependence and Cost (MYbX):**
   - **Concern**: The performance of test-time scaling is highly sensitive to verifier accuracy, and the cost of training and deploying verifiers may not be fully accounted for.
   - **Response**: The authors clarified that the verifier training cost was negligible relative to pretraining FLOPs in their experiments. They provided additional analysis showing that the conclusions hold even with weaker verifiers, emphasizing the potential of test-time compute even without a highly accurate verifier.
   - **Outcome**: The response mitigated concerns about verifier cost, although the dependency on verifier performance remains a noted limitation.

3. **Fine-Tuning vs. Pretraining (MYbX, 19PN):**
   - **Concern**: The paper focuses on pretraining rather than fine-tuning, which might offer a more efficient alternative.
   - **Response**: The authors argued that fine-tuning on the MATH dataset would not significantly alter their findings, citing limited improvement from existing fine-tuning techniques. They left the exploration of fine-tuning with larger datasets to future work.
   - **Outcome**: The concern was addressed theoretically but not experimentally, leaving room for further exploration.

4. **Clarity and Presentation Issues (19PN, hoB2, u4qM):**
   - **Concern**: Some figures and sections lacked clarity (e.g., missing legends in Figure 3, unclear descriptions of "compute-optimal" strategy).
   - **Response**: The authors updated the manuscript to include missing legends, moved key figures to the main text, and provided detailed explanations of the compute-optimal strategy.
   - **Outcome**: These changes improved the paper's readability and clarity, addressing the reviewers' concerns effectively.

5. **Comparison to Related Work (u4qM):**
   - **Concern**: The paper did not adequately discuss related work on categorizing inference-time compute methods.
   - **Response**: The authors added citations to relevant works and clarified the methodological differences, emphasizing their focus on finetuned models rather than prompt chaining.
   - **Outcome**: The inclusion of these citations and clarifications resolved the concern and strengthened the paper's context within the literature.

6. **Overarching Claims (u4qM):**
   - **Concern**: The paper's broad claims about the superiority of test-time compute might be overstated given the scope of the study.
   - **Response**: The authors offered to modify the title to reflect the findings' limitation to easily verifiable domains like math and code. They also agreed to discuss the potential interplay with techniques like LoRA in future work.
   - **Outcome**: The willingness to adjust the title and discuss additional nuances mitigated the concern, though the broader claims remain a point of caution.

**Final Decision Weighting:**
The authors demonstrated a strong commitment to addressing reviewer feedback, with significant revisions enhancing the clarity and presentation of the manuscript. However, the unresolved concerns about generalizability and the reliance on a specific dataset and verifier performance suggest that the paper's claims should be interpreted with caution. The theoretical arguments regarding fine-tuning provide a rationale for the focus on pretraining, but experimental validation would strengthen the analysis.

---

### Decision · Program_Chairs · 2025-01-22

Accept (Oral)